# A identification method for critical causes of lifting injuries based on topological potential

**Yingliu Yang**[1,2], **Lianghai Jin**[1,2,3]*

**1** College of Hydraulic & Environmental Engineering, Three Gorges University, Yichang Hubei, China, **2** Hubei Key Laboratory of Hydropower Engineering Construction and Management, Three Gorges University, Yichang Hubei, China, **3** Safety & Environment Science and Technology Ltd, Yichang Hubei, China

\* 202108590021106@ctgu.edu.cn

**Data Availability Statement:** The data has been uploaded to a public database. http://osf.io/wkmzg/files/osfstorage.

**Funding:** This work was supported by the National Natural Science Foundation of China (No.

## Abstract

In view of the deficiency that the traditional importance ranking method cannot be used to objectively and comprehensively evaluate the importance of the causes of hoisting injuries, an importance ranking method based on topological potential is proposed by using complex network theory and field theory in physics. First, the causes of 385 reported lifting injuries are divided into 36 independent causes at four levels through a systematic analysis approach, and the relationships among these causes are obtained through the Delphi method. Then, the accident causes are treated as nodes, and the relationships among the causes are used as edges to establish a network model of the causes of lifting accidents. The out-degree and in-degree topological potential of each node are calculated, and an importance ranking of lifting injuries causes is obtained. Finally, based on 11 evaluation indexes commonly used to assess node importance (node degree, betweenness centrality, etc.), the ability of the method proposed in this paper to effectively identify the key nodes in the cause network of lifting accidents is verified, and the conclusions can guide the safe implementation of lifting operations.

## Introduction

A crane is one of the most critical and expensive pieces of equipment on a construction site [1]. It performs the tasks of moving, lifting, and positioning heavy objects during construction operations. However, lifting operations are among the main causes of death in the construction process [2]. According to statistics, lifting operations cause up to one-third of deaths in the construction industry [3]. In addition, according to official data, lifting accidents in the construction industry in China in recent years are becoming increasingly common, Fig 1 shows the proportions of heavy accidents and lifting accidents in 2013–2019 (inconsistent data for 2020–2021) [4]. When a lifting accident occurs, it can injure workers and severely damages surrounding equipment and buildings, resulting in massive losses. For example, on July 5, 2020, a tower crane collapsed during a jacking operation in Area C of the Shuangchuang Industrial Park Project in Fenghuanghu District, Fengtai Economic Development Zone, Fengtai County, Anhui Province, killing five workers and causing a direct economic loss of nearly

52179136), but the funders had no role in study design, data collection and analysis, decision to publish, or preparation of the manuscript.

**Competing interests:** The authors declare that they have no known competing financial interests or personal relationships that could have appeared to influence the work reported in this paper.

1.348 million dollars (Huainan Emergency Management Bureau 2021); similarly, on May 17, 2018, a tower crane collapsed in Wuzhishan, Hainan, killing four people and causing a direct economic loss of approximately 0.931 million dollars (Department of Emergency Management of Hainan Province 2018).

The lifting operation construction environment is complex, the construction period is long, and it involves human factors, material factors, environmental factors, etc [5], is a multifaceted information interaction of complex system problems; human factors mainly include human unsafe behavior [6], which is the main cause of lifting accidents; the problem of management factors is the premise of the unsafe behavior of people and the unsafe state of things. The combined effect of multiple accident causes results in lifting accidents [7]; the safety state of lifting operations changes as human factors, material factors, environmental factors, and management factors change. The interaction between the causes of lifting injury accidents is high-order nonlinear, and the relationship between the causes of accidents is complex, which increases the complexity and difficulty of safety control of lifting operations and brings serious challenges to the government's public safety governance.

Hence, it is of great practical significance to analyze the occurrence mechanisms of lifting injuries, identify the key causes, improve the safety management efficiency of lifting

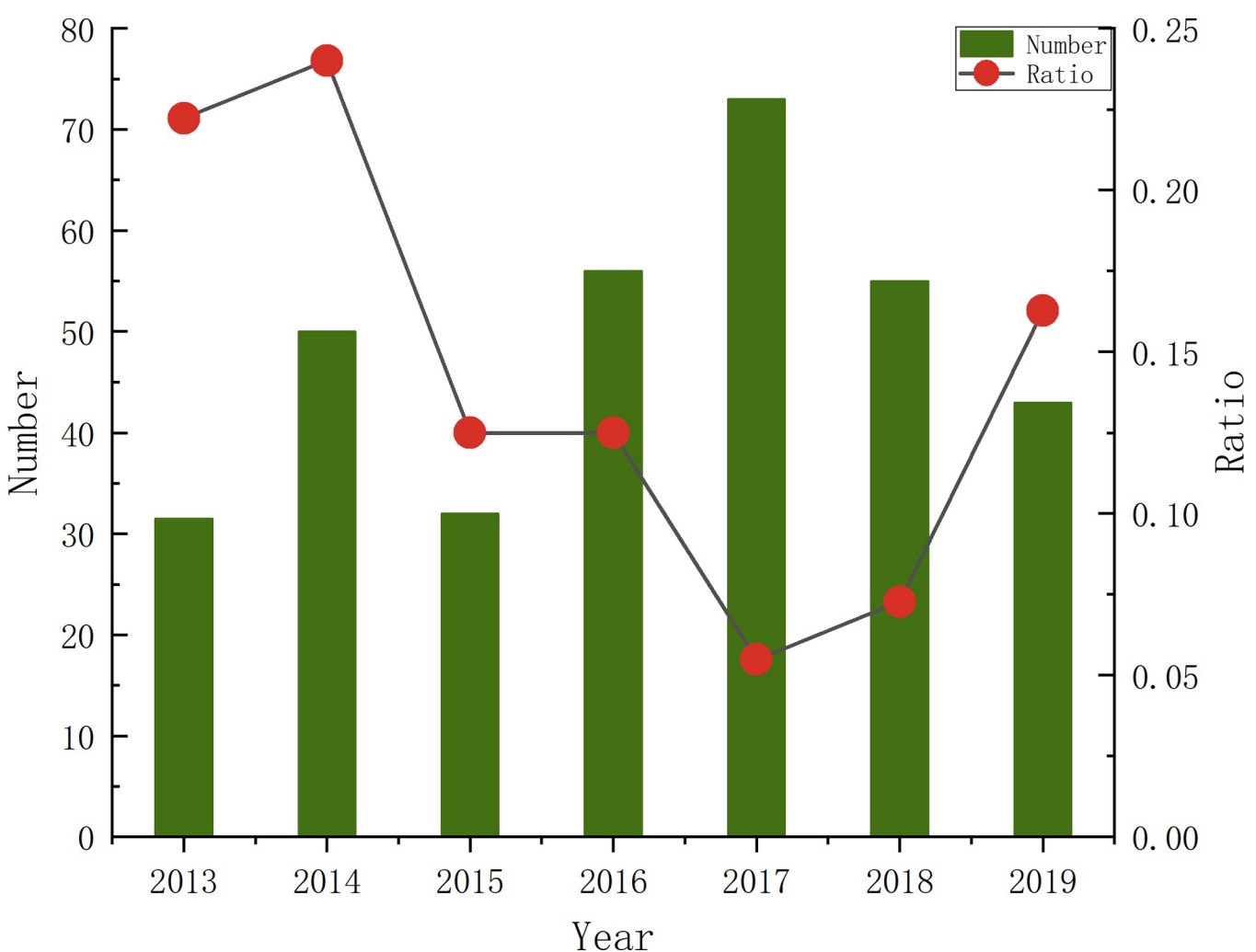

**Fig 1. Proportion of heavy accidents and lifting accidents.**

operations, and take effective measures to prevent or reduce the occurrence of lifting injuries [8]. In conclusion, systematic analyses of the causes and effects of lifting injuries can strengthen the application of accident cause theory in the construction field, and analyses of actual accident cases can provide a practical reference for the operation of tower cranes at construction sites [9].

Some scholars have performed extensive research on the factors that influence lifting operation safety and the causes of lifting injuries; this research includes accident analyses, construction site interviews [10], questionnaires [11], and other methods, and the results suggest that the main causes of lifting operation accidents are human error, equipment failure, and lack of management and training. The types of cranes involved in these accidents include mobile cranes [12], bridge cranes[13,14], gantry cranes [15], tower cranes, and ship cranes [16]. Furthermore, Tu [17] investigated the dependability of human factors, Lee [18] investigated the impact of equipment on the success of lifting operations, and Li [19] explored the effects of environmental factors. Furthermore, some researchers have investigated the safety factors involved in crane operation, installation, and disassembly. Notably, tower lifting accidents were divided into six levels using systematic thinking and a complex network; 34 disaster-causing factors and 10 accident types were identified, and 7 key factors and 3 key paths of tower lifting accidents in China were distinguished. Through systematic thinking and case analysis, the key causes of tower lifting accidents can be identified, such as worker errors, insufficient safety training, insufficient safety inspections, low safety awareness, and the poor management of safety engineers [20]. Through the Delphi method, the risk factors that affect the lifting operation were studied, and it was concluded that more attention should be given to the factors that highly influence site safety during tower crane work.

At present, the measurement of the critical causes of accidents is mainly based on frequency and complex network topology parameters [21–23], ignoring the role of potential field in the whole network of nodes, resulting in inaccurate evaluation results [24]. The emergence of topological potential theory can solve this problem. According to topological geometry, the cause network of lifting injury accidents is abstracted as a system composed of N nodes and their influence relations. The accident will form a potential field in the system. According to the topological distance between nodes, the role of accident cause is studied, and the key causes of lifting injury accidents are explored. Based on this, a prevention and control strategy for lifting injury accidents is proposed to improve the safety management efficiency of lifting operations. It is of great practical significance to take effective measures to prevent or reduce the occurrence of lifting injuries [25].

In summary, preliminary research has focused on determining the basic causes of lifting injuries, thus providing a good foundation for analyzing the causes of lifting injuries. However, little attention has been given to the key causes, and the current evaluation methods are limited, resulting in inaccurate evaluation results. On this basis, the concept of importance is used in this study to identify the key factors that lead to lifting injuries using a combined systematic thinking and topological potential theory approach. Then, the relationships among different causes determined to provide a reference for safety control and accident prevention during hoisting operations.

## Data collection

Firstly, a total of 385 valid accident reports from 2000 to 2020 were collected from the Safety Management Network (http://www.safehoo.com/Case/). Among them, 111 accident reports occurred in 2005–2021, accounting for 29% of the total accident reports, indicating that the safety form of hoisting operations in China is not optimistic. According to statistics, the 385

accidents caused a total of 667 deaths and 358 injuries, with an average of 1.73 deaths and 0.92 injuries per accident. The number of deaths caused by these accidents is 1.86 times the number of injuries, indicating that the severity of lifting injuries is high.

### Data analysis

Due to the limited data collected in this study and incomplete statistics, we can only make statistical analysis on the characteristics of 385 lifting injury accidents in the month and accident level. Among them, the accident without personal injury is separated from the general accident level and named "No Casualty Accident."

Making statistics on the accident grade of 385 lifting injuries, and the distribution is shown in Fig 2. As can be seen from the figure, there are 5 accidents without casualties, accounting for 1.3%, although there are no casualties, these accidents are often accompanied by huge economic losses. There are as many as 309 general accidents, accounting for 80.3%, and 60 major accidents, accounting for 15.6%, indicating that the damage caused by lifting injuries is relatively small, and in great cases, there are a few casualties. Besides, there were 8 heavy accidents, accounting for 2.1%, the most serious of which was the "4.13" crane overturning accident in Dongguan Dongjiangkou Prefabricated Component Factory, which caused 18 deaths and 33 injuries, with a direct economic loss of 18.61 million yuan. It was a major liability accident caused by extreme weather, inadequate maintenance and use of special equipment, unfulfilled safety production responsibility, and inadequate safety supervision. However, there were three extra serious accidents, accounting for 0.8%, the most serious of which was the collapse of a gantry crane in Shanghai Hudong Shipyard in 2001, which caused 36 deaths and 3 injuries, resulting in a direct economic loss of more than 80 million yuan. It was a major liability

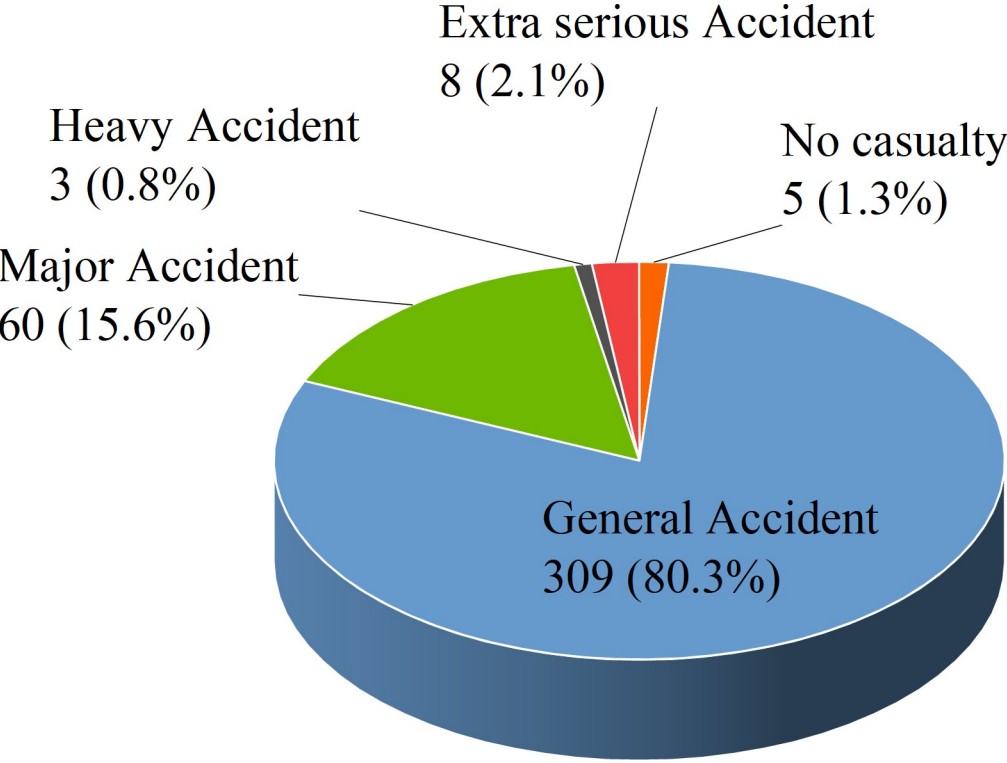

**Fig 2. Lifting injury accident grade distribution.**

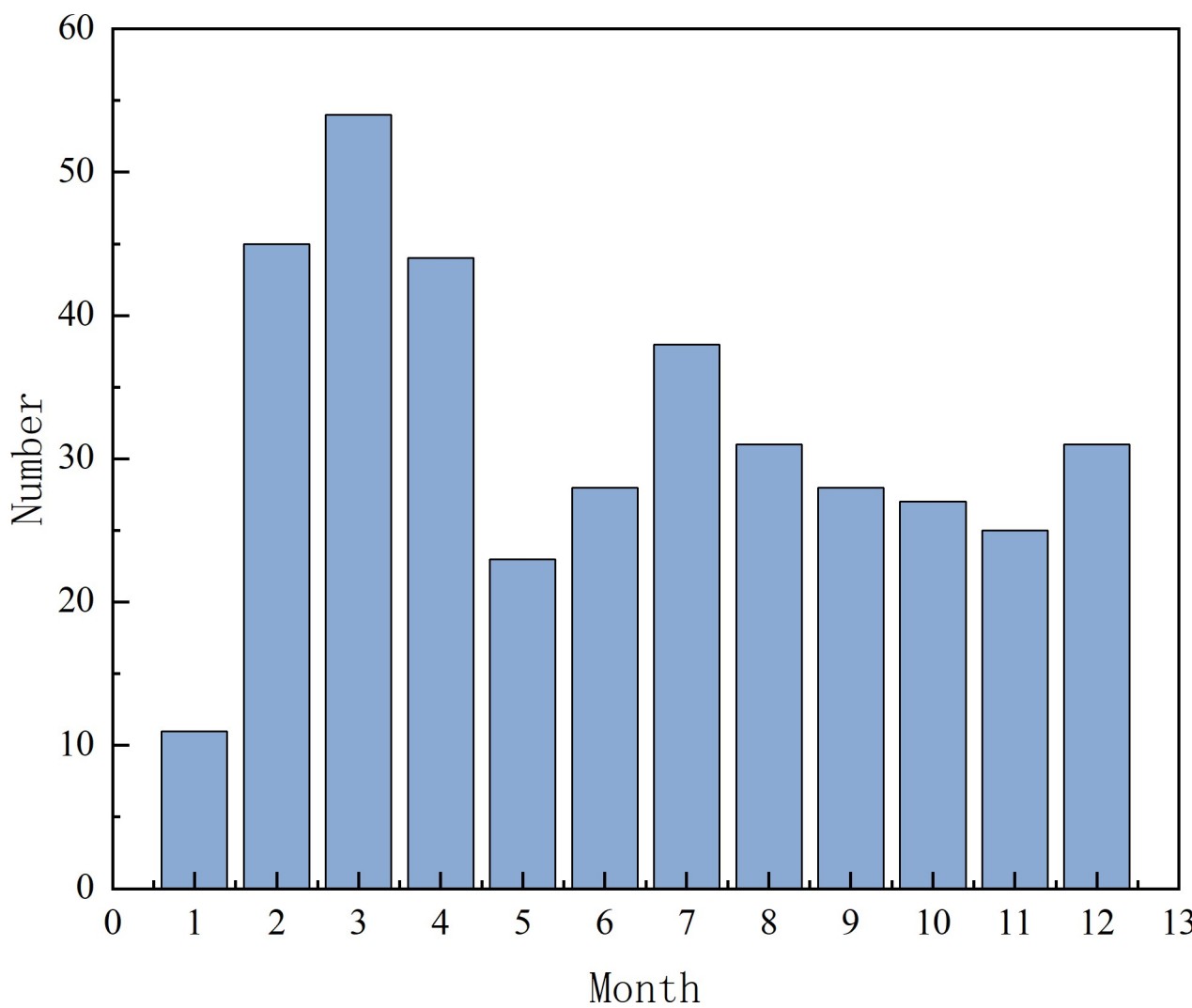

**Fig 3. Month distribution of heavy injury accidents.**

accident caused by an imperfect special hoisting construction scheme, illegal command, illegal operation, and inadequate on-site supervision during hoisting.

Fig 3 shows the results of monthly statistics of lifting injury accidents. It is implied that January has the least number of accidents, and May-June and August-November have fewer accidents, which are the months with better lifting operation. February-April has the largest number of accidents, and July and December have a large number of accidents, which are the most frequent months for lifting injuries. This is because June is China's statutory safety production month. During this period, all units pay close attention to safety management and control, and timely investigate potential safety hazards, effectively reducing the probability of hoisting injury accidents; July is a hot summer and December is a cold winter. At this time, the working environment is harsh, which increases the possibility of hoisting safety accidents. In January, during the Spring Festival in China, the story rate of lifting operations began to stop work one after another, which was relatively low; February-April is the month with high incidence of lifting accidents. At this time, as the lifting operation starts to resume work, the weather gets warmer, which has a great psychological impact on the operators.

### Classification of causes of lifting injury accidents

Then, the system theory was used to extract the accident causes from the accident reports, and after merging the causes with similar meanings and performing correlation tests, 36 independent causes were obtained in human, machine, environment, and management classes, as shown in Table 1. Finally, with the Delphi method, 15 experts in the field of lifting safety operations were invited to score the degree of influence among disaster-causing factors, and the scores ranged from 1 to 4. The larger the score was, the closer the connection between the two disaster-causing factors, and the answers of each expert were assigned the same weight. Since lifting activities occur during engineering practice, all the invited experts had more than 10 years of experience working at lifting construction sites, and they included 4 project managers, 6 operators, and 3 signalmen. In addition, 2 experts were university teachers who have been engaged in teaching and scientific research in construction safety management for more than 15 years. Table 2 shows an expert's rating of the impact of disaster-causing factors. By combining the scoring results of 15 experts, the relationship matrix for accident-causing factors was obtained.

## Network model of the causes of lifting accidents

### Establishment of a cause network for lifting accidents

A network model can describe the relationships among the factors the cause lifting accidents. Therefore, the proposed network model consists of two parts. First, the network nodes represent the causes of 36 lifting accidents, as listed in Table 1. A total of 36 nodes are established as the basis of the network model. Second, the connections between nodes, which are obtained with the Delphi method during the data collection phase, are formed. Ultimately, Gephi software is used to generate the corresponding network structure diagram of the causative factors, as shown in Fig 4. Then, we normalize the relation weights in the relation matrix.

### Measurement of the cause network for lifting accidents

**Topological potential theory.** According to topological geometry [26], we established a weighted undirected network $G = (V, L)$, where $V$ is the set of nodes and $L$ is the set of edges. The nodes themselves form a potential field in the network, which interacts with other nodes.

The topological potential algorithm for directed weighted networks was proposed by [27], and it divides node topological potential into out-degree topological potential and in-degree topological potential. The network with extension weighting is denoted as $E = (V, L, M, W)$, where $V$ is the set of nodes, $L$ is the set of directed edges, $M$ is the set of the attributes of nodes, and $W$ is the set of the weights of directed edges. In addition, the topological potential value of node $v_j$ at $v_i$ can be obtained with Eq (1), and the out-degree topological potential $\varphi_{out}(v_i)$ and in-degree topological potential $\varphi_{in}(vi)$ of node $v_i$ can be obtained with Eq (2) and Eq (3), respectively.

$$\varphi(v_j - v_i) = m_j e^{-(dw_{j-i}/\delta)^2} \tag{1}$$

$$\varphi_{out}(v_i) = \sum_{j=1}^{N} \left(m_j e^{-(dw_{j-i}/\delta)^2}\right) \tag{2}$$

$$\varphi_{in}(v_i) = \sum_{j=1}^{N} \left(m_j e^{-(dw_{j-i}/\delta)^2}\right) \tag{3}$$

**Table 1. Causes of lifting accidents.**

| Layer | Cause | Description |
|---|---|---|
| Human (*H*) | *H*1: Low safety awareness | Lack of safety education and training of personnel on site, such as command personnel standing in the crane swing radius during command lifting; |
| | *H*2: Lacking safety knowledge | Safety education of field personnel is not in place, resulting in a lack of relevant safety knowledge; |
| | *H*3: Operation violation | Operators who violate the relevant operating regulations, such as lifting the driver without authorization or signal workers who command lifting without authorization. |
| | *H*4: Illegal command | Field command personnel violate lifting operation command regulations, unauthorized command lifting operation |
| | *H*5: Risky operation | Operators master the relevant operating procedures, but the lack of safety knowledge or awareness makes them forced to operate. |
| | *H*6: Improper positioning | Personnel on site illegally enter dangerous areas of lifting operations, such as within the swing range of cranes or under lifting objects. |
| | *H*7: Operators working without certification | Operators do not have relevant qualifications to operate. |
| | *H*8: Inadequate use of PPE | Failure to wear or use protective equipment in accordance with the relevant provisions, such as aerial work without a seat belt, entering the scene without a helmet, etc; |
| | *H*9: Lack of professional skills | The vocational skills of field workers do not meet the relevant operational requirements, such as inexperienced lifting operations by drivers. |
| | *H*10: Workers with bad physical or mental conditions | Workers' physical and mental conditions, such as drinking or being injured but not fully healed, do not meet operating standards. |
| | *H*11: Overloaded lifting | The weight of the lifting operation exceeds the safe range. |
| | *H*12: Improper selection of the lifting point | Improper selection of lifting points, such as when workers arbitrarily decide lifting points. |
| Management (*P*) | *P*1: Lack of safety responsibility | Each responsible unit's safety responsibility is merely a formality and is not in place |
| | *P*2: Inadequate safety supervision | No person in charge or responsible person is in place, responsibility is not clear, and multi-sectoral or multi-person management is tantamount to no management; |
| | *P*3: Inadequate construction technical disclosure | There is no construction technology disclosure or demonstration in place, it is not thorough, it is not targeted, it is weak, and it cannot be used in construction operations; |
| | *P*4: Inadequate maintenance and management of special equipment | Such as the crane is unqualified or structurally aging, lack of maintenance; |
| | *P*5: Inadequate safety education and training | Safety education and training have devolved into a formality, with no goal of achieving a unified scientific and technological norm, standardized operations, or preventing employees from identifying work site hazards; |
| | *P*6: Inadequate on-site supervision | The construction site did not arrange full-time supervisors as required or have them in place; |
| | *P*7: Inadequate special construction scheme | No special construction plan is prepared, or the integrity and feasibility of a special construction plan do not conform to; |
| | *P*8: Inadequate hidden danger investigation | There was no investigation or discovery of a safety hazard on the construction site; |
| | *P*9: No emergency plan | Production and operation units fail to Eqte emergency plans as required; |
| | *P*10: Unlicensed equipment | Equipment not licensed for use at the construction site; |
| | *P*11: Inadequate contracting administration | The management system of a construction general contracting enterprise is not perfect, and effective construction contracting management is not carried out; |
| | *P*12: Inadequate safety regulations for lifting operations | Lifting operation safety regulations are not designed to meet requirements. |
| Machine (*M*) | *M*1: Unstable foundation of the tower crane | Crane collapse due to insufficient foundation strength during lifting operations; |
| | *M*2: Failure or unreliable structural connections | Such as the lifting boom connection bolt is not reliable or falls, resulting in collapse during the lifting process and causing an accident; |
| | *M*3: Damaged mechanical or functional components | The crane torque limiter fails or malfunctions, causing a lifting injury accident; |
| | *M*4: Failure of safety devices | Such as crane lifting limiter switch damage and amplitude limiter failure; |
| | *M*5: Unreliable auxiliary assembly | Such as a crane lifting limiter failure can result in an overload lifting accident. |
| | *M*6: Failure of lifted object | For example, the rope strength is not enough, lifting process leads to rope fracture, hook fracture, slippage, etc; |
| | *M*7: Design or manufacturing defects in special equipment | Crane problems in the design stage or manufacturing process problems. |

(*Continued*)

**Table 1.** (Continued)

| Layer | Cause | Description |
|---|---|---|
| Environment (E) | E1: Adverse climate conditions | Weather conditions are not suitable for lifting operations, such as typhoons, sudden heavy rain, high temperatures, high winds, and other bad weather; |
| | E2: Interference from other operations | Two or more construction operations are carried out at the same time or in the same space, causing accidents; |
| | E3: Adverse work environment | The site construction environment does not meet the requirements, such as disorderly stacking of materials, and the flatness of the site does not meet the requirements; |
| | E4: Inadequate warning signs | The site did not set up warning signs in accordance with the regulations, and the warning effect was not enough; |
| | E5: High schedule pressure | Compression or insufficiency of construction period, resulting in continuous overwork of workers. |

where $dw_{j-i}$ is the distance between nodes in the network considering the relevant weights and $\sigma$ is the influence of the potential field, which is used to control the influence range of nodes and can be optimized according to the topological potential quotient of nodes. Let the shortest path from node $v_j$ to $v_i$ pass through edges $l_1, l_2, \ldots, l_h$ in turn, with a total of $h$ breaks; $d_r$ is the length of section $r$ along this path, and $w_r$ is the edge weight of this section. Therefore, we can establish Eq (4):

$$dw_{j-i} = \sum_{r=1}^{h} \frac{d_r}{w_r} \tag{4}$$

Generally, the influence of the mass of a node and the length of edges is ignored, and these values are both assumed to be 1. Notably, in a directed network, the out-degree topological potential $\varphi_{out}(v_i)$ and in-degree topological potential $\varphi_{in}(v_i)$ of a node $v_i$ represent the influence of the node on other neighboring nodes and the degree of influence of other nodes, respectively.

A topological index can describe the interactions among nodes and reflect the position differences of nodes and the importance of their attributes. The value of the topological potential quotient can reflect the uncertainty degree of node position differences [28]; when the topological potentials of all nodes in the network are equal, the uncertainty degree of node position differences is at a maximum, and the corresponding potential quotient is the largest. When the topological potentials of nodes are unequal, the uncertainty and the potential quotient reach minimum values. In addition, as the number of influential factors gradually increases, the potential quotient first decreases and then increases; that is, it reaches a maximum value at

**Table 2. The top 10 of network out-degree and in-degree topological potential of lifting injury accidents.**

| No. | Cause | $\varphi_{out}(v_i)$ | Cause | $\varphi_{in}(v_i)$ |
|---|---|---|---|---|
| 1 | P1 | 28.962 | M3 | 18.714 |
| 2 | P2 | 28.506 | H6 | 16.940 |
| 3 | P11 | 20.251 | H5 | 15.036 |
| 4 | P12 | 16.608 | H8 | 15.003 |
| 5 | H7 | 11.459 | M6 | 14.279 |
| 6 | P5 | 10.042 | H3 | 13.245 |
| 7 | H2 | 9.613 | H11 | 11.325 |
| 8 | H1 | 8.394 | H12 | 11.007 |
| 9 | P6 | 8.277 | H4 | 9.219 |
| 10 | H3 | 6.525 | M2 | 6.901 |

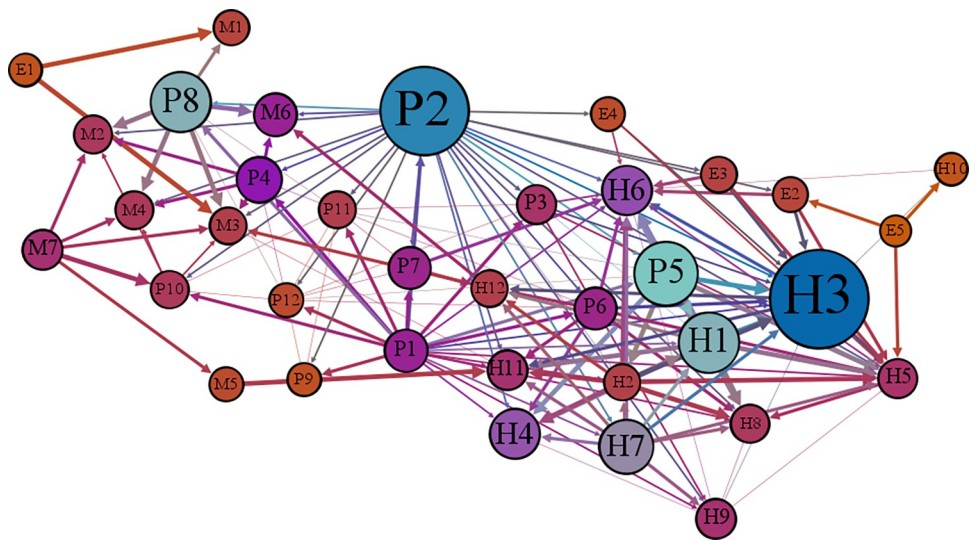

**Fig 4. Network structure diagram of the factors that cause lifting accidents.**

both ends of an edge, and there is a minimum value in the middle. The minimum value of the potential quotient corresponds to the optimal influential factor; the corresponding topological potential distribution of nodes is highly heterogeneous, and the uncertainty is minimized. It is worth noting that if there are disconnected node pairs separated by an infinite distance, the potential quotient cannot reach the true minimum value; thus, $\sigma \rightarrow \infty$ [29]. Specifically, the potential quotient can be obtained as follows:

$$H(\delta) = -\sum_{i}^{n} \frac{\varphi(v_i)}{Z} \ln \frac{\varphi(v_i)}{Z} \tag{5}$$

where $Z = \sum_{i}^{N} \varphi(v_i)$ is a standardized parameter.

According to the $3\sigma$ rule for the Gaussian potential function and the weight distance Eq for edges, the influence range $\lambda$ of each node is the neighborhood with a radius of approximately $3\sigma/\sqrt{2}$ centered on that node [30]; that is, all nodes with $dw_{j-i} \leq \lambda$ are within the range of influence.

### Calculation of the network parameters for lifting injury causes

**Calculation of optimal influence factor.**  Usually, the optimal influential factor in a complex network is obtained based on potential entropy calculations. The optimal factor $\sigma_{opt}$ = 1.886 is obtained with Eq (5) in this study. Fig 5 shows the changes in the corresponding out-degree and in-degree topological potentials of the crane accident cause network for different influential factors. When the optimal factor increases from 0.47 to 2.36, the difference in node topological potential values due to the different positions of nodes in the network becomes increasingly obvious. However, when $\sigma_{opt}$ = 1.886, the discrimination level is the highest, so $\sigma_{opt}$ = 1.886 should be used to calculate the two-dimensional topological potential.

**Topological trend of cause of lifting injury accident.**  In general, the optimal influence factor in complex networks is obtained by calculating the potential entropy. The optimal impact factor, $\sigma_{opt}$ = 1.886, of the causal network is calculated by Eq (5). The optimal impact factor of lifting operation accident causation network topology potential of the top 10 causes is shown in Table 2. The node importance calculation results are shown in Fig 5.

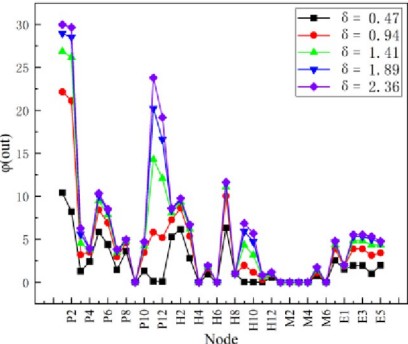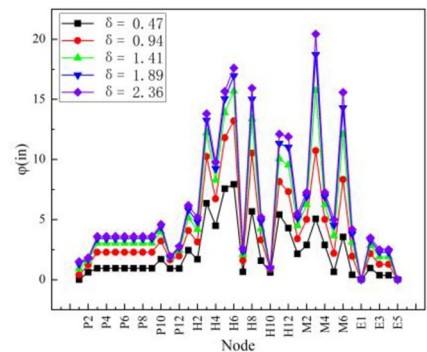

**Fig 5. The out-degree and in-degree topological potentials for different influential factors.**

It can be seen from the table that: 1) the top 10 accident causations of out-degree topological potential only include two levels of human and management, and on the whole, the management level is higher than the human level, indicating that the management level has a greater impact on the remaining accident causations; 2) The top 10 accident causations of in-degree topological potential only include human and machine levels, and in general, the number of human level causations is more than that of machine level, indicating that human unsafe behavior is more likely to be affected by other accident causations. 3) The absence of the environmental level in the top ten accident causes of out-degree topological potential and in-degree topological potential indicates that the environmental level factors have little influence on the other accident causes and are not affected by the other accident causes.

The results of the node importance calculations for the network of lifting accident causes based on the optimal influential factors is shown in Fig 6. Notably, 1) node $P_1$ displays the highest influence and out-degree topological potential, and node $M_3$ exhibits the greatest influence and in-degree topological potential, corresponding to the mechanical or functional parts that are damaged; 2) nodes $P_2$ and $P_{11}$ rank second and third in topological potential, respectively, and the corresponding causes are inadequate safety supervision and inadequate contract management, which are important intermediary causes of hoisting accidents; and 3) nodes $H_6$ and $H_5$ are ranked fourth and fifth in topological potential, respectively, and the corresponding causes are improper lifting stations and risky operations, which are affected by other processes.

**Causative attribute discovery of serious injury accident.** To further research the attributes of each node, based on the optimal influential factor $\sigma$, the out-degree and in-degree

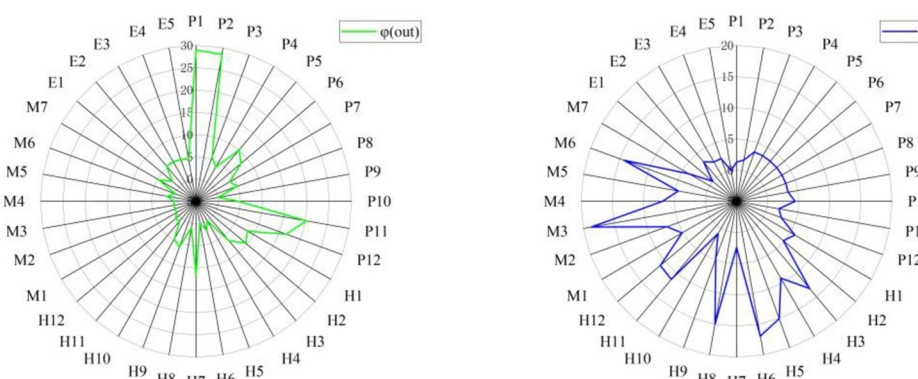

**Fig 6. The out-degree and in-degree topological potentials with σopt = 1.886.**

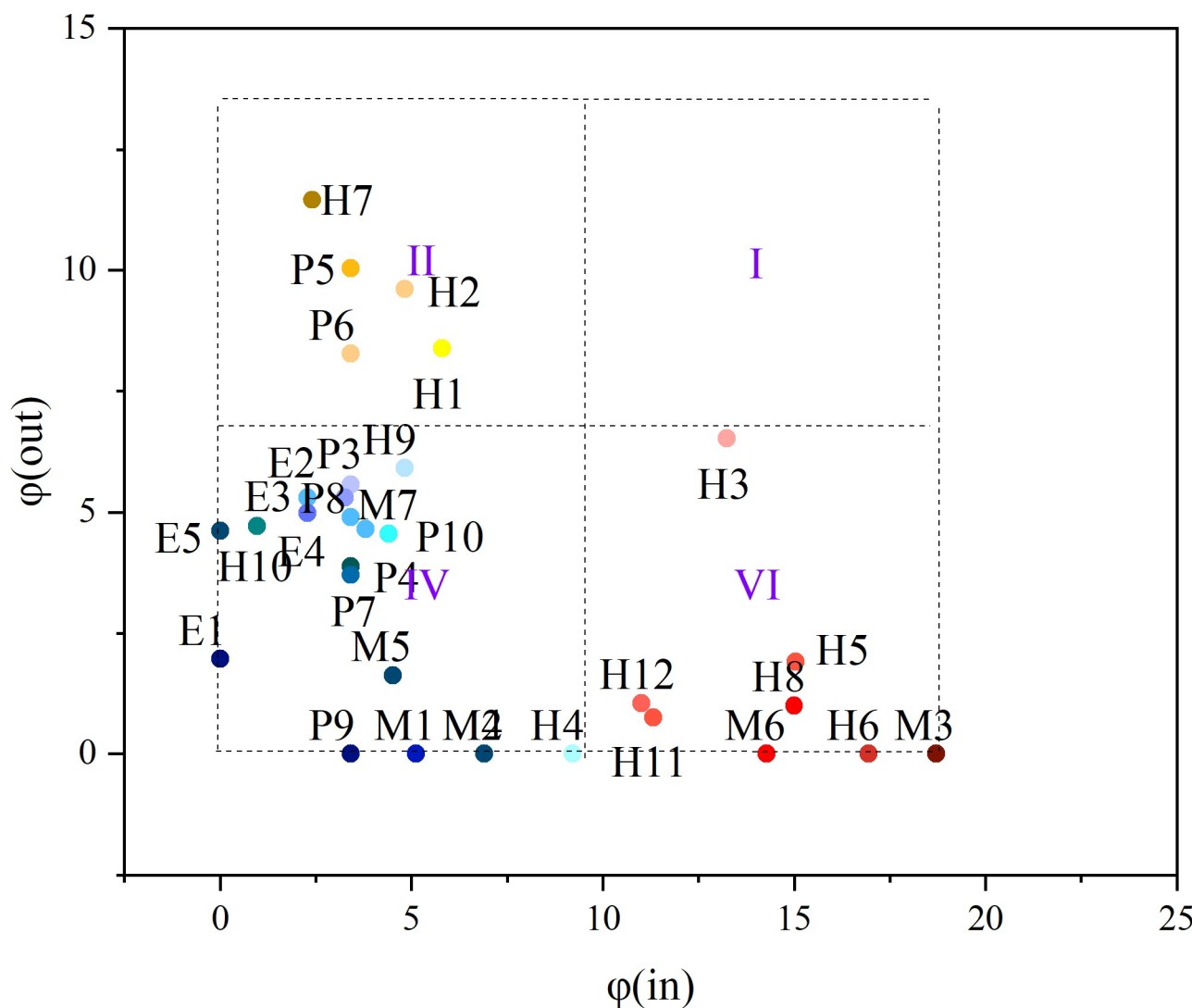

**Fig 7. Diagram of the two-dimensional topological potential distribution.**

topological potential values of each node are calculated, and plotting the distribution of the two-dimensional topological potential of nodes, and the area is divided into four parts: I, II, IV, and VI (Fig 7) [31]. The darker the color of the node in each area is, the higher the degree to which the node is associated with the attributes of that area.

Area I: This type of node has large topological potentials for the out-degree and in-degree and plays a connecting role in the network;

Area II: The out-degree topological potential of this type of node is high, and the in-degree topological potential is low;

Area VI: The out-degree topological potential of this type of node is low, and the in-degree topological potential is high;

Area IV: The topological potential of this type of node is low.

According to the corresponding definitions of the areas in Fig 4, it can be concluded that 1) most nodes have small out-degree and in-degree topological potentials and are mainly related to hoisting machinery and the environment, indicating that these factors are influenced little by other nodes and have little influence on other nodes; 2) *P5*, *P6*, *H1*, *H2*, and *H7* are located

in Area II, which is characterized by a high out-degree topological potential and a low in-degree topological potential; the corresponding factors, which are inadequate safety education and training, inadequate on-site supervisors, low safety awareness, lack of safety knowledge, and operators working without certification, are affected little by other factors but have a high impact on other factors. Consequently, these factors should be the focus of safety control during crane operations; 3) *H3*, *H5*, *H8*, *H11*, *H12*, *M3*, and *M6* are located in Area VI, corresponding to violation operations, risky operations, inadequate use of PPE, overloaded lifting, improper selection of lifting points, damaged mechanical or functional components, and failure of the object being lifted. These factors have little influence on other factors, but they are all greatly influenced by other factors, making them the direct causes of lifting accidents.

## Comparison of topological potential calculations

To evaluate the effect of the out-degree and in-degree topological potentials on node importance, 12 classic indexes, including the weighted out-degree, weighted in-degree, betweenness centrality, clustering coefficient, nearness centrality, and others, are selected for comparison. The results are shown in Tables 3 and 4.

From Tables 3 and 4, it can be concluded that 1) the top 5 nodes in terms of out-degree topological potential and in-degree topological potential also display high ranks for other important metrics; for example, *P1*, which ranks first in out-degree topological potential, ranks in the top 5 for seven other important metrics; 2) the out-degree topological potential of the degree metric is the closest to that of the out-degree, but the former yields higher accuracy by optimizing differences in the weighting degree and enhancing the distinction of importance among nodes; and 3) similarly, the in-degree topological potential of the degree metric is the closest to that of the in-degree, but the former yields higher accuracy.

In summary, the out-degree topological potential and in-degree topological potential of nodes calculated with the topological potential method reflect the importance of nodes. The attributes of nodes can be identified from the diagram of the two-dimensional topological potential distribution, and the conclusions from the analysis of topological values are realistic based on actual conditions.

## Conclusions

Based on the concept of node importance, the topological structure of the nodes that cause lifting accidents is investigated by using a method that combines systematic thinking and topological potential theory to evaluate the importance of nodes in a critical cause network for lifting accidents. The main conclusions of this study are as follows:

According to 385 accident reports for lifting injuries, the causes of lifting injuries are divided into four classes, namely, human, machine, environment, and management. Then, causes are further decomposed into 36 independent accident causes related to the safety management of lifting operations.

A topological network structure is established, and it includes 36 nodes and 161 connecting edges. Each node represents an accident cause, and the weights of connecting edges depend on the interactions among causes. Additionally, the size of nodes reflects the frequency of the cause in the 385 accident reports. The importance of accident causes can be measured based on the out-degree topological potential and in-degree topological potential of the network.

After network analysis and calculations, the key causes of lifting injuries are determined, such as those that have the greatest influence on other causes but are affected little by other causes (*P5*, *P6*, *H1*, *H2*, and *H7*) and those that have little influence on other causes but are greatly influenced by other causes (*H3*, *H5*, *H8*, *H11*, *H12*, *M3*, and *M6*).

                                    

**Table 3. Rankings for the top 5 nodes in terms of out-degree topological potential.**

| Factor | Φout | Φin | Out-degree | In-degree | Degree | Weighted out-degree | Weighted in-degree | Weighted overall degree | Betweenness centrality | Clustering | Eigencentrality | Closeness centrality | Eccentricity | Harmonic closeness centrality |
|---|---|---|---|---|---|---|---|---|---|---|---|---|---|---|
| P1 | 1 | 25 | 3 | 11 | 2 | 1 | 20 | 2 | 11 | 20 | 21 | 5 | 2 | 5 |
| P2 | 2 | 24 | 1 | 10 | 1 | 2 | 17 | 1 | 3 | 23 | 18 | 3 | 2 | 2 |
| P11 | 3 | 23 | 3 | 10 | 5 | 11 | 16 | 16 | 18 | 15 | 18 | 11 | 2 | 11 |
| P12 | 4 | 20 | 5 | 9 | 7 | 14 | 15 | 18 | 20 | 12 | 16 | 12 | 2 | 12 |
| H7 | 5 | 21 | 4 | 10 | 6 | 4 | 17 | 8 | 19 | 7 | 18 | 2 | 3 | 2 |

**Table 4. Rankings for the top 5 nodes in terms of in-degree topological potential.**

| Factor | Φout | Φin | Out-degree | In-degree | Degree | Weighted out-degree | Weighted in-degree | Weighted overall degree | Betweenness centrality | Clustering | Eigencentrality | Closeness centrality | Eccentricity | Harmonic closeness centrality |
|---|---|---|---|---|---|---|---|---|---|---|---|---|---|---|
| M3 | 28 | 1 | 12 | 5 | 9 | 19 | 5 | 13 | 20 | 25 | 4 | 17 | 5 | 16 |
| H6 | 28 | 2 | 12 | 1 | 5 | 19 | 1 | 5 | 20 | 18 | 1 | 17 | 5 | 16 |
| H5 | 23 | 3 | 10 | 2 | 4 | 15 | 2 | 4 | 7 | 14 | 2 | 1 | 4 | 1 |
| H8 | 26 | 4 | 12 | 4 | 8 | 19 | 4 | 10 | 20 | 4 | 3 | 17 | 5 | 16 |
| M6 | 28 | 5 | 12 | 5 | 12 | 19 | 10 | 20 | 20 | 21 | 5 | 17 | 5 | 16 |

The topological potential analysis method proposed in this paper can be effectively used to identify the key causes of accidents during in lifting operations with a network approach, and the conclusions obtained provide a reference for the development of hazard identification, accident prevention, accident rescue, and accident investigation in lifting operations.

## Author Contributions

**Conceptualization:** Yingliu Yang.

**Data curation:** Yingliu Yang.

**Formal analysis:** Yingliu Yang.

**Funding acquisition:** Lianghai Jin.

**Methodology:** Yingliu Yang.

**Resources:** Lianghai Jin.

**Software:** Yingliu Yang.

**Supervision:** Yingliu Yang, Lianghai Jin.

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
