## [Decision Letter · Decision Letter 0]

14 Dec 2022

PONE-D-22-27160Research on Key Factor Identification Method of Lifting Injury Cause Network Based on Topological PotentialPLOS ONE

Dear Dr. Yang,

Thank you for submitting your manuscript to PLOS ONE. After careful consideration, we feel that it has merit but does not fully meet PLOS ONE’s publication criteria as it currently stands. Therefore, we invite you to submit a revised version of the manuscript that addresses the points raised during the review process.

ACADEMIC EDITOR: Please further highlight the contribution of this article through comparison and other ways. 

We look forward to receiving your revised manuscript.

Kind regards,

Zilin Gao, Ph.D

Academic Editor

PLOS ONE

Journal Requirements:

“This work was supported by the National Natural Science Foundation of China (No. 52179136)”

“This work was supported by the National Natural Science Foundation of China (No. 52179136).”

“This work was supported by the National Natural Science Foundation of China (No. 52179136)”

6. One of the noted authors is a group or consortium “lianghai jin”. In addition to naming the author group, please list the individual authors and affiliations within this group in the acknowledgments section of your manuscript. Please also indicate clearly a lead author for this group along with a contact email address.

7. We note that you have stated that you will provide repository information for your data at acceptance. Should your manuscript be accepted for publication, we will hold it until you provide the relevant accession numbers or DOIs necessary to access your data. If you wish to make changes to your Data Availability statement, please describe these changes in your cover letter and we will update your Data Availability statement to reflect the information you provide.

8. In your Data Availability statement, you have not specified where the minimal data set underlying the results described in your manuscript can be found. PLOS defines a study's minimal data set as the underlying data used to reach the conclusions drawn in the manuscript and any additional data required to replicate the reported study findings in their entirety. All PLOS journals require that the minimal data set be made fully available. For more information about our data policy, please see http://journals.plos.org/plosone/s/data-availability.

Reviewers' comments:

Reviewer's Responses to Questions

**Comments to the Author**

1. Is the manuscript technically sound, and do the data support the conclusions?

Reviewer #1: Partly

Reviewer #2: Yes

2. Has the statistical analysis been performed appropriately and rigorously? 

Reviewer #1: No

Reviewer #2: Yes

3. Have the authors made all data underlying the findings in their manuscript fully available?

Reviewer #1: No

Reviewer #2: Yes

4. Is the manuscript presented in an intelligible fashion and written in standard English?

Reviewer #1: Yes

Reviewer #2: No

5. Review Comments to the Author

Reviewer #1: The intent of this paper is to analyze the key factor of lifting injury cause through topological potential of complex networks and the systematic thinking. The nodes and edges are the accident causes and the influence relationship between the causes respectively. All I can see is the idea of using a complexity science perspective to solve a practical problem, and that idea might be useful. However, in terms of innovation and theoretical contribution, this article is not suitable for publication in Plos One.

Firstly, there are many ways to obtain key factors of lifting injury cause, but this paper cannot illustrate the advantages of the complex network approach and practical application value of the results.

In addition, the Delphi method is a traditional expert decision making method that is highly subjective. The link relationships of network nodes are likely to change due to subjectivity.

Moreover, the topological potential of complex networks is methods for measuring the importance of network nodes. This theory of measuring the importance of nodes has been proposed for a long time. This paper only uses this method to conduct an empirical study and does not make clear how to improve this method in theory.

Reviewer #2: 1.It is noted that your manuscript needs careful editing by someone with expertise in technical English editing paying particular attention to English grammar, spelling, and sentence structure so that the goals and results of the study are clear to the reader.

2. The title should be reconsidered, especially the description“Lifting Injury Cause Network”, i think this expression is somewhat repetitive and suggest modifying it to “Lifting Accident Cause Network”.

3.In section 2 “Data collection “, what is the situation of heavy injury accidents in recent years ? i think the statistical description of data samples should be added.

4.In the first paragraph of the introduction, just describes the trend of heavy injury accidents, but I think it is not intuitive, it is recommended to increase the statistical or trend graph of lifting injury accidents.

5.In the penultimate paragraph of the introduction, the limitations of current research on lifting injury accidents and the innovative description of this study are not enough, and it is recommended to increase additional description and analysis.

6.Table 1, What is the author 's definition of the cause of each accident ? I think your manuscript should increase the description of the cause of each lifting injury accident.

7.The structure of Section 3.3 is somewhat unreasonable. I think it should be divided into several sections, and the manuscript needs to added some research discussions about the human, machine, environmental, and management levels.

6. PLOS authors have the option to publish the peer review history of their article (what does this mean?). If published, this will include your full peer review and any attached files.

Reviewer #1: No

Reviewer #2: **Yes: **Jianlan Zhou

---

## [Author Response · Author response to Decision Letter 0]

18 Jan 2023

Dear Editor and Reviewers:

We are so appreciated for processing the manuscript (Manuscript ID: PONE-D-22-27160) entitled “Research on Key Factor Identification Method of Lifting Injury Cause Network Based on Topological Potential” by Yingliu Yang, Lianghai Jin*. We are truly grateful to the reviewers’ comments and recommendations on improving the manuscript, which indeed help us greatly to improve the quality of our paper. Based on these comments and suggestions, we have carefully considered and addressed all comments in the revised manuscript.

Please find replies (line number referred to revised manuscript) to reviewers’ comments. If any questions remain, please let us know. We are looking forward to hearing good news from you! Thanks for your attention and patience!

Best regards,

Yingliu Yang

Hubei Key Laboratory of Disaster Prevention and Mitigation, China Three Gorges University, Yichang 443002, China

Hubei Key Laboratory of Hydropower Engineering Construction and Management, China Three Gorges University, Yichang 443002, China

E-mail: yangyingliu2023@163.com

Below you will find our point-by-point responses to the reviewers’ comments/ questions.

Appendix: Response to reviewers' comments and specific remarks

Response to reviewer #1:

Comment 1: The intent of this paper is to analyze the key factor of lifting injury cause through topological potential of complex networks and the systematic thinking. The nodes and edges are the accident causes and the influence relationship between the causes respectively. All I can see is the idea of using a complexity science perspective to solve a practical problem, and that idea might be useful. However, in terms of innovation and theoretical contribution, this article is not suitable for publication in Plos One.

Firstly, there are many ways to obtain key factors of lifting injury cause, but this paper cannot illustrate the advantages of the complex network approach and practical application value of the results.

In addition, the Delphi method is a traditional expert decision making method that is highly subjective. The link relationships of network nodes are likely to change due to subjectivity.

Moreover, the topological potential of complex networks is methods for measuring the importance of network nodes. This theory of measuring the importance of nodes has been proposed for a long time. This paper only uses this method to conduct an empirical study and does not make clear how to improve this method in theory..

Reply: Thanks for your valuable comments. Firstly, based on a large number of accident reports, this paper uses text mining technology to obtain the causes of lifting injury accidents, which has certain objectivity. Mining the causes of key accidents through topological potential theory can be used as the focus of lifting operation safety control. Secondly, the use of the Delphi method to obtain the relationship between accident causes has a certain initiative, but also has a certain significance. In future research, we will adopt a more objective method ; finally, complex networks are widely used, so the evaluation results are also highly reliable. Therefore, this study uses this method to verify the results. Your opinion is very good, and we will conduct corresponding research in the future.

Response to reviewer #2:

Comment 1: It is noted that your manuscript needs careful editing by someone with expertise in technical English editing paying particular attention to English grammar, spelling, and sentence structure so that the goals and results of the study are clear to the reader.

Reply: Thanks for your valuable comments. We have revised the full text as requested. Please see the revised version for details.

Comment 2: The title should be reconsidered, especially the description“Lifting Injury Cause Network”, i think this expression is somewhat repetitive and suggest modifying it to “Lifting Accident Cause Network”.

Reply: Thanks for your valuable suggestion. We have revised the title of the paper.

Original text: Research on Key Factor Identification Method of Lifting Injury Cause Network Based on Topological Potential.

Revised version: A Identification Method for Critical Causes of Lifting Injuries based on Topological Potential.(Lines:1-3)

Comment 3: In section 2 “Data collection “, what is the situation of heavy injury accidents in recent years ? i think the statistical description of data samples should be added.

Reply: Thanks for your valuable comments. First of all, we have carried out statistical analysis of the data used in this paper according to the requirements, and then analyzed the characteristics of the accident in the month.(Lines:95-137)

Comment 4: In the first paragraph of the introduction, just describes the trend of heavy injury accidents, but I think it is not intuitive, it is recommended to increase the statistical or trend graph of lifting injury accidents.

Reply: Thanks for your valuable comments, we have added the 2013-2019 lifting accident trends as requested.

Original text: In addition, according to official data, lifting accidents in China 's construction industry in recent years are not optimistic and show an upward trend (Zhang et al., 2019).

Revised version: In addition, according to official data, lifting accidents in the construction industry in China in recent years are becoming increasingly common , Fig 1 shows the proportions of heavy accidents and lifting accidents in 2013-2019 (inconsistent data for 2020-2021)[4].(Lines:30-32)

Fig 1. Proportion of heavy accidents and lifting accidents

Comment 5: In the penultimate paragraph of the introduction, the limitations of current research on lifting injury accidents and the innovative description of this study are not enough, and it is recommended to increase additional description and analysis.

Reply: Thanks very much for your valuable suggestion. We have revised the introduction as requested.

Original text: At present, the measurement of key nodes in complex networks is mainly based on topological characteristic parameters, including degree centrality (Bonacich., 1972), intermediate centrality (Freeman., 1977), proximity centrality (Sabidussi., 1966), feature vector centrality (Newman., 2006), etc. These parameters can effectively measure the influence between nodes themselves and connected nodes, but ignore the influence of other nodes in the whole network (Kong et al., 2017; Wang et al., 2019).

Revised version: At present, the measurement of the critical causes of accidents is mainly based on frequency and complex network topology parameters [15-18], ignoring the role of potential field in the whole network of nodes, resulting in inaccurate evaluation results [19, 20] [14]. The emergence of topological potential theory can solve this problem. According to topological geometry, the cause network of lifting injury accidents is abstracted as a system composed of N nodes and their influence relations. The accident will form a potential field in the system. According to the topological distance between nodes, the role of accident cause is studied, and the key causes of lifting injury accidents are explored. Based on this, a prevention and control strategy for lifting injury accidents is proposed to improve the safety management efficiency of lifting operations. It is of great practical significance to take effective measures to prevent or reduce the occurrence of lifting injuries.(Lines:76-85)

Comment 6: Table 1, What is the author 's definition of the cause of each accident ? I think your manuscript should increase the description of the cause of each lifting injury accident.

Reply: Thanks for your valuable comments, we have in accordance with the requirements of the table a serious injury accident cause description.

Revised version: A Identification Method for Critical Causes of Lifting Injuries based on Topological Potential.(Line:152)

Layer Cause Description

Human (H) H1: Low safety awareness Lack of safety education and training of personnel on site, such as command personnel standing in the crane swing radius during command lifting;

 H2: Lacking safety knowledge Safety education of field personnel is not in place, resulting in a lack of relevant safety knowledge;

 H3: Operation violation Operators who violate the relevant operating regulations, such as lifting the driver without authorization or signal workers who command lifting without authorization. 

 H4: Illegal command Field command personnel violate lifting operation command regulations, unauthorized command lifting operation

 H5: Risky operation Operators master the relevant operating procedures, but the lack of safety knowledge or awareness makes them forced to operate.

 H6: Improper positioning Personnel on site illegally enter dangerous areas of lifting operations, such as within the swing range of cranes or under lifting objects.

 H7: Operators working without certification Operators do not have relevant qualifications to operate.

 H8: Inadequate use of PPE Failure to wear or use protective equipment in accordance with the relevant provisions, such as aerial work without a seat belt, entering the scene without a helmet, etc;

 H9: Lack of professional skills The vocational skills of field workers do not meet the relevant operational requirements, such as inexperienced lifting operations by drivers. 

 H10: Workers with bad physical or mental conditions Workers' physical and mental conditions, such as drinking or being injured but not fully healed, do not meet operating standards. 

 H11: Overloaded lifting The weight of the lifting operation exceeds the safe range. 

 H12: Improper selection of the lifting point Improper selection of lifting points, such as when workers arbitrarily decide lifting points.

Management (P) P1: Lack of safety responsibility Each responsible unit's safety responsibility is merely a formality and is not in place

 P2: Inadequate safety supervision No person in charge or responsible person is in place, responsibility is not clear, and multi-sectoral or multi-person management is tantamount to no management;

 P3: Inadequate construction technical disclosure There is no construction technology disclosure or demonstration in place, it is not thorough, it is not targeted, it is weak, and it cannot be used in construction operations;

 P4: Inadequate maintenance and management of special equipment Such as the crane is unqualified or structurally aging, lack of maintenance;

 P5: Inadequate safety education and training Safety education and training have devolved into a formality, with no goal of achieving a unified scientific and technological norm, standardized operations, or preventing employees from identifying work site hazards;

 P6: Inadequate on-site supervision The construction site did not arrange full-time supervisors as required or have them in place;

 P7: Inadequate special construction scheme No special construction plan is prepared, or the integrity and feasibility of a special construction plan do not conform to；

 P8: Inadequate hidden danger investigation There was no investigation or discovery of a safety hazard on the construction site;

 P9: No emergency plan Production and operation units fail to Eqte emergency plans as required;

 P10: Unlicensed equipment Equipment not licensed for use at the construction site;

 P11: Inadequate contracting administration The management system of a construction general contracting enterprise is not perfect, and effective construction contracting management is not carried out;

 P12: Inadequate safety regulations for lifting operations Lifting operation safety regulations are not designed to meet requirements. 

Machine (M) M1: Unstable foundation of the tower crane Crane collapse due to insufficient foundation strength during lifting operations;

 M2: Failure or unreliable structural connections Such as the lifting boom connection bolt is not reliable or falls, resulting in collapse during the lifting process and causing an accident;

 M3: Damaged mechanical or functional components The crane torque limiter fails or malfunctions, causing a lifting injury accident;

 M4: Failure of safety devices Such as crane lifting limiter switch damage and amplitude limiter failure;

 M5: Unreliable auxiliary assembly Such as a crane lifting limiter failure can result in an overload lifting accident.

 M6: Failure of lifted object For example, the rope strength is not enough, lifting process leads to rope fracture, hook fracture, slippage, etc;

 M7: Design or manufacturing defects in special equipment Crane problems in the design stage or manufacturing process problems.

Environment (E) E1: Adverse climate conditions Weather conditions are not suitable for lifting operations, such as typhoons, sudden heavy rain, high temperatures, high winds, and other bad weather;

 E2: Interference from other operations Two or more construction operations are carried out at the same time or in the same space, causing accidents;

 E3: Adverse work environment The site construction environment does not meet the requirements, such as disorderly stacking of materials, and the flatness of the site does not meet the requirements;

 E4: Inadequate warning signs The site did not set up warning signs in accordance with the regulations, and the warning effect was not enough;

 E5: High schedule pressure Compression or insufficiency of construction period, resulting in continuous overwork of workers.

Comment 7: The structure of Section 3.3 is somewhat unreasonable. I think it should be divided into several sections, and the manuscript needs to added some research discussions about the human, machine, environmental, and management levels.

Reply: Thanks for your valuable comments, according to the opinion, we list the topology potential value of larger nodes, and based on human, machine, environment, management four levels of research.(Lines:218-233)

Revised version: In general, the optimal influence factor in complex networks is obtained by calculating the potential entropy. The optimal impact factor, opt = 1.886, of the causal network is calculated by Eq ([Disp-formula pone.0283144.e005]) . The optimal impact factor of lifting operation accident causation network topology potential of the top 10 causes is shown in Table 2. The node importance calculation results are shown in Fig 5.

Table 2 The top 10 of network out-degree and in-degree topological potential of lifting injury accidents

No. Cause φout(vi) Cause φin(vi)

1 P1 28.962 M3 18.714

2 P2 28.506 H6 16.940

3 P11 20.251 H5 15.036

4 P12 16.608 H8 15.003

5 H7 11.459 M6 14.279

6 P5 10.042 H3 13.245

7 H2 9.613 H11 11.325

8 H1 8.394 H12 11.007

9 P6 8.277 H4 9.219

10 H3 6.525 M2 6.901

It can be seen from the table that : 1 ) the top 10 accident causations of out-degree topological potential only include two levels of human and management, and on the whole, the management level is higher than the human level, indicating that the management level has a greater impact on the remaining accident causations ; 2 ) The top 10 accident causations of in-degree topological potential only include human and machine levels, and in general, the number of human level causations is more than that of machine level, indicating that human unsafe behavior is more likely to be affected by other accident causations. 3) The absence of the environmental level in the top ten accident causes of out-degree topological potential and in-degree topological potential indicates that the environmental level factors have little influence on the other accident causes and are not affected by the other accident causes.

---

## [Decision Letter · Decision Letter 1]

3 Mar 2023

A identification method for critical causes of lifting injuries based on topological potential

PONE-D-22-27160R1

Dear Dr. Yang,

We’re pleased to inform you that your manuscript has been judged scientifically suitable for publication and will be formally accepted for publication once it meets all outstanding technical requirements.

Kind regards,

Zilin Gao, Ph.D

Academic Editor

PLOS ONE

Additional Editor Comments (optional):

Reviewers' comments:

Reviewer's Responses to Questions

**Comments to the Author**

1. If the authors have adequately addressed your comments raised in a previous round of review and you feel that this manuscript is now acceptable for publication, you may indicate that here to bypass the “Comments to the Author” section, enter your conflict of interest statement in the “Confidential to Editor” section, and submit your "Accept" recommendation.

Reviewer #1: All comments have been addressed

Reviewer #2: All comments have been addressed

2. Is the manuscript technically sound, and do the data support the conclusions?

Reviewer #1: Yes

Reviewer #2: Yes

3. Has the statistical analysis been performed appropriately and rigorously? 

Reviewer #1: Yes

Reviewer #2: Yes

4. Have the authors made all data underlying the findings in their manuscript fully available?

Reviewer #1: Yes

Reviewer #2: Yes

5. Is the manuscript presented in an intelligible fashion and written in standard English?

Reviewer #1: Yes

Reviewer #2: Yes

6. Review Comments to the Author

Reviewer #1: (No Response)

Reviewer #2: The authors have replied my concerns perfectly. I would like this paper to be accepted as soon as possible.

7. PLOS authors have the option to publish the peer review history of their article (what does this mean?). If published, this will include your full peer review and any attached files.

Reviewer #1: No

Reviewer #2: No

---

## [Editor Report · Acceptance letter]

12 Mar 2023

PONE-D-22-27160R1 

A identification method for critical causes of lifting injuries based on topological potential 

Dear Dr. Yang:

I'm pleased to inform you that your manuscript has been deemed suitable for publication in PLOS ONE. Congratulations! Your manuscript is now with our production department. 

Kind regards, 

on behalf of

Dr. Zilin Gao 

Academic Editor

PLOS ONE